# Integrated Transcriptome Analysis Identified Key Expansin Genes Associated with Wheat Cell Wall, Grain Weight and Yield

**DOI:** 10.3390/plants12152868

**Published:** 2023-08-04

**Authors:** Juan P. Mira, Anita Arenas-M, Daniel F. Calderini, Javier Canales

**Affiliations:** 1Instituto de Bioquímica y Microbiología, Facultad de Ciencias, Universidad Austral de Chile, Valdivia 5110566, Chile; jpmiraw@gmail.com (J.P.M.); ana.arenas@uach.cl (A.A.-M.); 2ANID-Millennium Science Initiative Program-Millennium Institute for Integrative Biology (iBio), Santiago 8331150, Chile; 3Plant Production and Plant Protection Institute, Faculty of Agricultural Sciences, Universidad Austral de Chile, Valdivia 5110566, Chile

**Keywords:** wheat grain development, expansin genes, cell wall modification, gene co-expression network, *Triticum aestivum*

## Abstract

This research elucidates the dynamic expression of expansin genes during the wheat grain (*Triticum aestivum* L.) development process using comprehensive meta-analysis and experimental validation. We leveraged RNA-seq data from multiple public databases, applying stringent criteria for selection, and identified 60,852 differentially expressed genes across developmental stages. From this pool, 28,558 DEGs were found to exhibit significant temporal regulation in at least two different datasets and were enriched for processes integral to grain development such as carbohydrate metabolism and cell wall organization. Notably, 30% of the 241 known expansin genes showed differential expression during grain growth. Hierarchical clustering and expression level analysis revealed temporal regulation and distinct contributions of expansin subfamilies during the early stages of grain development. Further analysis using co-expression networks underscored the significance of expansin genes, revealing their substantial co-expression with genes involved in cell wall modification. Finally, qPCR validation and grain morphological analysis under field conditions indicated a significant negative correlation between the expression of select expansin genes, and grain size and weight. This study illuminates the potential role of expansin genes in wheat grain development and provides new avenues for targeted genetic improvements in wheat.

## 1. Introduction

Wheat, as the most widely cultivated cereal, plays a crucial role in global food security by providing approximately one-fifth of the world’s total caloric intake (https://www.fao.org/faostat/en/#home (accessed on 20 June 2023)). However, the escalating threat of global warming necessitates improvements in wheat production to meet rising global demands, given that current production rates are insufficient [1,2]. Therefore, the need arises to optimize wheat production, particularly focusing on the grain growth phases.

Grain development comprises three distinct phases—grain growth, grain filling, and desiccation—involving a series of cellular events within different grain tissues: the endosperm, maternal tissues, and embryo [3]. These stages coincide with substantial alterations in gene expression linked to essential processes that drive development, tissue specificity, and the function of transcription factors [4]. Transcriptomic time-series studies on developing whole caryopses of wheat have illuminated a plethora of differentially expressed genes (DEGs), associated with cell division, starch biosynthesis, and hormone biosynthesis, thereby indicating dynamic and intricate gene regulation throughout the process [5,6,7].

Grain yield in wheat predominantly depends on two factors: the number of grains per square meter and grain weight, the latter of which directly correlates with grain volume [8]. Grain weight is influenced by grain dimensions (width, length, and height), and its potential is primarily determined during a brief window of grain development [9,10,11]. Concurrently, strong associations have been established between grain weight and floral carpel size at anthesis in both sunflower and wheat crops [9,12]. This reinforces the hypothesis that the maternal tissues forming the grain’s outer layers, the pericarp, influence potential grain weight by physically constraining expansion, thereby determining grain volume [9,13,14].

Concomitantly, pericarp growth is highly active during the early stages of grain filling, with grain size modifications, especially length, defined during the growth phase [8]. A strong correlation exists between these alterations and final grain weight. Notably, grain elongation is primarily driven by the rate of cell elongation [15]. Consequently, the processes governing cell wall expansion are critical for potential grain weight. For cell growth to occur, the cell wall must reorganize and extend, transitioning from rigidity to a viscoelastic or relaxed state, which facilitates cell expansion [16].

At the molecular level, the expression of certain protein factors, such as expansins, plays a paramount role in this cell expansion process, crucial for defining grain size [17]. Expansins, ubiquitous across plants, can soften and extend the cell wall, while working in tandem with xyloglucan-endotransglycosylase/hydrolases (XTHs), which modify the cell wall [16]. Expansins comprise a superfamily of proteins that are essential for stress relaxation and cell wall extension in plants [18]. Classified into four subfamilies: α expansin (EXPA), β expansin (EXPB), expansin-like A (EXLA), and expansin-like B (EXLB), these proteins are typically 250–275 amino acids in length, characterized by two specific domains [19]. Multiple members of the EXPA and EXPB subfamilies have demonstrated in vitro expansin activity, specifically the ability to induce rapid extension or stress relaxation of isolated cell walls placed under uniaxial tension [19]. The biomechanical properties of the cell wall, mediated by expansin action, regulate plant development and growth by controlling cell elongation and primordia specification/outgrowth [20].

Extensive research on expansins from various plant species has revealed their influence on nearly all growth phases and their potential impact on plant–biotic/abiotic stress relationships [17]. However, only a handful of expansins have been associated with crop improvement or yield enhancement. For instance, a recent study highlighted the potential to increase grain weight without altering grain numbers through the ectopic expression of a wheat root expansin, *TaEXPA-6*, in endosperm and pericarp tissues during the initial stages of grain development [21]. Despite the wheat genome encoding 241 expansin genes divided into three subfamilies, EXPA, EXPB, and EXLA [22], only a small fraction has been functionally characterized, leaving a substantial knowledge gap about the specific roles and expression patterns of expansins during grain growth [21,23,24,25].

Our study aims to identify key expansin genes that play essential roles in cell wall loosening, stress relaxation, and cell expansion, which are fundamental to grain growth. We propose that the use of bioinformatics tools to analyze publicly available RNA-seq data will yield robust, verifiable results about gene expression during wheat grain development. The knowledge gained from this research could greatly benefit future wheat breeding programs, primarily focusing on enhancing grain yield—an imperative goal considering the increasing global food demand.

## 2. Results

### 2.1. Data Collection and Identification of DEGs during Wheat Grain Development

In this study, a comprehensive meta-analysis was performed on publicly available RNA-seq data with the aim of identifying cell wall-associated genes that exhibit differential expression during wheat grain development. The analysis was specifically focused on bread wheat (*Triticum aestivum* L.) to avoid potential complexities associated with the diverse ploidy levels inherent in different wheat species. The selected studies met critical criteria: they were peer reviewed for quality assurance, employed consistent sequencing platforms to maintain comparability, used whole-grain samples, and captured at least three temporal stages to delineate expression dynamics throughout grain development.

We selected three datasets from the GEO and SRA databases [6,26,27], comprising 100 samples across seven distinct stages of grain development (Table 1). Over 2.1 billion raw reads from these datasets were pseudoaligned to the reference transcriptome of *Triticum aestivum* cv. Chinese Spring (IWGSC RefSeq v1.1) using kallisto software [28] (Appendix A). In particular, approximately 60% of these reads were successfully aligned, with an average of 12.7 million reads per sample.

The DESeq2 R package [29] was employed to investigate DEGs across all temporal pairings within each dataset, applying a false discovery rate (FDR) threshold of 0.05 and a two-fold change cutoff. In total, we identified 60,852 DEGs spanning the developmental stages in all three datasets (Appendix A). The first dataset yielded 46,833 DEGs, primarily observed between 15 and 35 days after anthesis (DAA). The second dataset revealed 10,686 DEGs, predominantly from 9 to 25 DAA. The third dataset contained 40,142 DEGs, with the greatest number found when comparing 5 DAA samples with those from 10, 15, and 20 DAA.

### 2.2. Robust Identification of DEGs during Grain Development

We established a set of robustly regulated DEGs by intersecting unique genes from each dataset. There were 8341 genes consistently present across all three datasets and 28,558 genes shared between at least two datasets (Figure 1B,C, Appendix A). Considering a good balance between consistency and information, we decided to select these genes that showed significant temporal regulation in at least two different datasets for further analyses.

To ascertain whether the selected genes provide insight into the grain development process, we conducted an enrichment analysis of gene ontology terms associated with the 28,558 robust temporally responsive genes during wheat grain development. As depicted in Figure 2, the top five over-represented biological function GO terms are linked to well-known processes in grain development such as carbohydrate metabolism (adjusted *p*-value = 9.04 × 10${}^{-47}$, 1128 genes [6,7]), nucleosome organization (adjusted *p*-value = 1.03 × 10${}^{-30}$, 177 genes [30]), and cell wall organization and biosynthesis (adjusted *p*-value = 4.30 × 10${}^{-19}$, 310 genes [3,7]). The molecular functions and cellular component GO terms align with these primary functions. For example, ‘nucleosome’ and ‘extracellular region’ are among the most over-represented GO terms in the cellular component domain (Figure 2). These findings suggest that the selected genes are strongly associated with grain development, including genes involved in cell wall biosynthesis.

To gain further insights into the metabolic pathways associated with the differentially expressed genes during wheat grain development, we conducted KEGG pathway analysis. This revealed enrichment for several key pathways related to metabolism, biosynthesis, and photosynthesis (Figure 2B).

Notably, pathways linked to carbon fixation, glycolysis/gluconeogenesis, and the pentose phosphate shunt were significantly enriched, suggesting active metabolic processes related to cell wall precursor synthesis. Further, enrichment of ribosome biogenesis and DNA replication pathways reflects the active protein synthesis and cell division occurring during grain growth. Overall, the enriched KEGG pathways align with known biological processes integral to wheat grain development, supporting the idea that the differentially expressed gene set captures key activities associated with grain growth.

### 2.3. Temporal Dynamics and Subfamily Distribution of Expansin Genes during Grain Development

From a total of 241 expansin-encoding genes identified in the wheat genome [22], our comprehensive meta-analysis revealed 72 expansin genes that displayed differential expression over time during grain development in at least two different datasets. After eliminating functional redundancy attributed to ploidy, we distinguished 33 unique expansin genes spread across three subfamilies. Specifically, the TaEXPA subfamily had 10 unique genes (22 when counting homoeologs), the TaEXPB subfamily contained 17 (with 36 when homoeologs are included), and the TaEXLA subfamily comprised 6 unique genes (or 14 when including homoeologs) (refer to Figure 3A). Particularly, the TaEXLA subfamily was notably over-represented in our dataset (14 out of 16 genes, with a 3.3 fold-enrichment, *p*-value < 0.01), which suggests its potential significance in wheat grain growth.

To explore the temporal patterns of expansin expression during grain development, we employed hierarchical clustering using the meta-analysis RNA-seq data. To simplify data visualization, we averaged the expression of all samples at the same time point for each gene, a method that was extended to closely spaced time points, for example, 9 and 10 DAA. Our analysis discerned at least three distinct clusters within each expansin subfamily (Figure 3). A majority of the expansin genes, irrespective of their subfamily, exhibited a significant temporal decrease in expression from day 5 to 15 DAA, subsequently maintaining low expression levels thereafter. Despite this general pattern, we identified two subclusters: one demonstrating a rapid decrease at 9–10 DAA (cluster 2, Figure 3B,C), and another illustrating a later decrease between 9–10 DAA and 15 DAA (cluster 3, Figure 3B,C). These observations advance our understanding of the temporal regulation of expansin gene expression during grain development, implying a coordinated expression within this pivotal gene family for cell wall growth and remodeling.

Analyzing normalized expression levels, we aimed to discern which expansins might significantly contribute to the early stages of grain growth. As illustrated in Figure 3, the expansin gene with the highest expression on day 5 DAA was *TaEXPB1-A*, alongside its homoeologs *TaEXPB1-B*, *TaEXPB1-D*. Notably, another member of the EXPB subfamily with its three homoeologs was among the ten most expressed expansins at 5 DAA (*TaEXPB2*). Moreover, we detected two representatives from the EXLA subfamily, *TaEXLA3-B* and *TaEXLA5-D*, and two members of the EXPA subfamily, *TaEXPA9-B* and *TaEXPA33-A*, among the top 10. These findings deepen our understanding of the functional implications of different expansin subfamilies in the initial stages of grain development.

### 2.4. Gene Co-Expression Network Analysis Unveils Expansin Genes Associated with Cell Wall Development in Wheat Grains

Our initial investigation yielded 28,558 genes showing significant temporal regulation during wheat grain development (Appendix A). Using the weighted gene co-expression network analysis (WGCNA) method from the Bionero R package [33], we constructed a gene co-expression network, aiming to identify gene clusters associated with cell wall development and discerning which expansins genes are concurrently expressed with these specific clusters. The WGCNA enabled the classification of the identified genes into 28 discrete co-expression modules (Appendix A), with over half the genes concentrated within the top five modules: M1 to M5 (Figure 4A).

We streamlined the co-expression network by illustrating interactions with a Pearson correlation of 0.95 or higher, resulting in a network comprising 2793 genes and 10,002 interactions (Figure 4A). Notably, the larger modules, M1 and M2, displayed connections with smaller modules, including M3–M9 and M6, indicating potential functional interrelationships. An over-representation analysis suggested that M1 and M3 shared four GO terms associated with the cell wall, while M2 and M6 shared seven GO terms related to metabolism and signaling.

We pinpointed co-expression modules over-represented with cell wall-related GO terms and identified only two such modules, M1 and M3 (Appendix A). The distribution analysis revealed that the majority of expansins (75%) were located in the module M1, associated with the cell wall, while M3, another cell wall-related module, contained a fewer number of expansin genes (Figure 4B).

M1 genes were predominantly expressed on 5 DAA, with a sharp decline in expression on 9–10 DAA, and stabilization thereafter from 15 DAA (Figure 5A). The biological functions most over-represented in this module were “nucleosome assembly” and “photosynthesis”, along with several cell wall-related terms. Importantly, of the 345 cell wall-associated genes annotated in the wheat genome, 138 were found in Module 1, indicating a considerable enrichment of this biological process in this cluster.

In contrast, M3 was characterized by the enrichment of the GO terms “carbohydrate metabolic process” (183 genes), “starch biosynthetic process”, and “energy reserve metabolic process”, suggesting its role in grain filling. Among the ten most over-represented biological processes, two were cell wall-related: “cellulose metabolic process” (16 genes) and “cell wall organization” (29 genes). However, gene expression within this module did not diminish until 20 DAA, unlike M1 (Figure 5B). Collectively, these results suggest that distinct functional processes associated with cell wall development could operate at specific stages of grain development.

Our InterPro domain functional annotation analysis of genes associated with the commonly enriched GO term “cell wall organization or biogenesis” in both M1 and M3 modules revealed distinct domain distributions. Notably, the M1 module was predominantly characterized by the XTH (xyloglucan endotransglucosylase/hydrolases) functional domain (25%), while M3 was characterized by the cellulose synthase domain (19%) (Figure 5C), suggesting a differential temporal regulation between enzymes involved in cell wall modification and those engaged in biogenesis.

In summary, most expansins demonstrated a decreasing expression pattern from 5 DAA, mirroring the expression pattern of other cell wall genes. Furthermore, we detected a small subset of expansins maintaining consistent expression levels during the first 20 DAA, correlating with the expression of cellulose synthase genes. This temporal pattern underscores the potential role of these expansins in wheat grain cell expansion and cell wall organization.

### 2.5. Differential Expression Patterns of Expansin Genes in Developing Grains Analyzed via qPCR and Their Correlation with Grain Morphology and Weight

In order to examine the expression patterns of expansin genes linked to cell wall-associated modules obtained in our meta-analysis, we conducted an experiment under ambient conditions during the 2021 season. We gathered grain samples at seven distinct stages of development: 5, 10, 15, 20, 25, 31, 35, and 40 DAA (Figure 6). Six expansin genes, two from each subfamily (EXPA: *TaEXPA1-A*, *TaEXPA3-B1*; EXPAB: *TaEXPB2-B*, *TaEXPB7-B*; EXLA: *TaEXLA2-D*, *TaEXLA3-B*), were chosen for validation through quantitative PCR (qPCR) in these samples. The selection was guided by consistency in the meta-analysis results, prioritizing expansin genes that showed differential expression across numerous samples. Our primary focus was on verifying the expression of expansins from the M1 module, which featured the highest number of cell wall genes. Moreover, we also analyzed the expression of an expansin gene from the M3 module (*TaEXPA3-B1*, Appendix A).

As shown in Figure 6, the expression of expansins *TaEXPA1-A*, *TaEXPB2-B*, and *TaEXLA-3B* significantly declined starting from 20 DAA. Before day 20, two distinct expression patterns were discernible: expansin *TaEXPB2-B* showed a notable decrease at each time point from 5 DAA to 20 DAA, while expansins *TaEXPA1-A* and *TaEXLA-3B* did not exhibit any significant decrease in their expression during this period. This latter pattern was similarly observed for expansins *TaEXPA3-B1*, *TaEXPB7-B*, and *TaEXLA2-D* (Appendix A), suggesting that most expansins maintain high expression levels during the grain’s exponential growth phase and subsequently decrease from 15 DAA.

To investigate if there is a correlation between the expression of selected expansins, and grain size and weight during their developmental stages, we assessed parameters such as grain length, width, area, and dry weight at the same stages of development as in the qPCR analysis. The expansins *TaEXPA1-A* and *TaEXPB2-B* demonstrated a significantly negative correlation (*p*-value < 0.01, R < −0.8) with these grain growth parameters, whereas *TaEXLA-3B* did not exhibit any significant correlation (Figure 7). Due to sample size constraints, we were unable to perform correlation analysis between expansin expression and grain growth parameters separately for each developmental stage. Further research with increased sampling at each stage will enable a stage-specific correlation analysis, which may provide more nuanced insights into these relationships. Similarly, two other expansins from the A and B subfamilies that were analyzed by qPCR, also showed negative correlations with grain weight and width (*TaEXPA3-B1* and *TaEXPB7-B*; see Appendix A). However, the other analyzed representative EXLAs, *TaEXLA2-D* and *TaEXLA3-B*, did not demonstrate any significant correlation with these traits (see Appendix A).

## 3. Discussion

Expansins are plant proteins that weaken the cell wall and participate in cell enlargement and other processes requiring cell wall modification, such as stress response [16,20]. In the wheat genome, 241 expansins have been identified [22], only a subset of which have been functionally characterized [21,23,24,25]. Despite the substantial presence of expansins in the wheat genome and their integral role in cell expansion, the comprehensive repertoire of specific expansins expressed during grain growth remains largely uncharacterized. Meta-transcriptomic analysis facilitates the identification of essential genes whose expression is independent of other variables, such as developmental stage or growth conditions [39,40,41]. Consequently, we conducted a meta-analysis of transcriptomics using three sets of publicly available RNA-seq data to establish which expansins are active during grain expansion and which are coexpressed with cell wall marker genes.

### 3.1. Identification of Temporally Regulated Expansins during Grain Growth

In our analysis, we identified 28,558 genes that are temporally regulated during grain development across at least two different data sets (Figure 1C). Functional enrichment analysis revealed that this robust gene set is indicative of the main processes known during grain growth, such as carbohydrate metabolism [6,7], nucleosome modification [30], and cell wall modification [3,7].

Having confirmed that our gene set accurately represents grain development, we proceeded to search for genes encoding expansins in wheat genome [22]. We discovered 72 expansins that are temporally regulated during wheat grain growth, corresponding to 33 unique genes after eliminating functional redundancy due to ploidy. Thus, we were able to rule out about 70% of the wheat genome expansins whose expression does not vary significantly during grain growth or are not expressed in this tissue. Specifically, the DESeq2 expression filter indicated 74 EXPA family expansins were not expressed in grain samples. Further, 11 EXPA family expansins lacked significant temporal regulation during grain development. Similarly, 42 EXPB family expansins were not expressed in grain based on the filter, and 14 others did not exhibit temporal regulation. Therefore, a considerable number of expansins do not appear to be active or have dynamic expression in the grain context. Elucidating the specific expansins relevant to grain facilitates more targeted functional characterization and understanding potential roles in grain growth.

However, the number of detected expansins is still quite high, considering, for example, that Arabidopsis has a total of 36 expansins in its genome [19]. The large number of wheat expansins is likely related to the distinctive cell wall composition of grasses compared with eudicots and most monocot groups [19,42]. These cell walls contain high levels of heteroxylans and relatively smaller amounts of heteromannans, pectic polysaccharides, and xyloglucans. Certain grasses and cereals also contain (1,3;1,4)-β-glucans, which are not widely distributed outside the Poaceae [43,44].

### 3.2. Functional Implications of Identified Expansins

Given the levels and timing of expression, we propose that a subset of the 72 identified expansins may play a pivotal role in wheat grain growth. The grain growth phase, characterized by cell wall initiation and pronounced mitotic division, typically unfolds within a 1–10 DAA window [45]. This phase is specifically associated with cellularization between 3 and 6 DAA [3,45]. Accordingly, our investigation centered on expansins exhibiting peak expression at 5 DAA, which aligns with cell wall initiation. 

Notably, *TaEXPB1*, including its three homologs A, B, and D, exhibited the most pronounced expression, followed by *TaEXPB2*. Intriguingly, previous research has reported that the overexpression of the *TaEXPB1* gene in Arabidopsis results in rapid root elongation, early bolting, and increases in leaf number, rosette diameter, and stem length [23]. These results suggests that *TaEXPB1* may play a functionally significant role in grain growth in wheat.

In the case of the EXPA subfamily, the genes *TaEXPA9*, *TaEXPA33*, and *TaEXPA1* displayed the highest expression during the early stages of grain growth. Interestingly, overexpression of *TaEXPA1* in Arabidopsis has been demonstrated to increase germination rates and yield larger cotyledons and rosette leaves [24]. These prior findings imply that the expansins identified through our meta-analysis strategy could be promising candidates for enhancing yield in wheat.

Moreover, we discovered that the TaEXLA subfamily was the most abundantly represented among expansins in our dataset of robustly regulated genes during wheat grain expansion. Alpha and beta expansins have been experimentally demonstrated to induce cell wall loosening [46,47], but scant information is available regarding the EXLA and EXLB family genes [19,48]. Despite structural similarities to other expansins and possessing signal peptides that target them to the cell wall, their exact function is still speculative [19]. Notably, non-flowering plants lack EXLA and EXLB genes [19], suggesting a potential association of these expansin families with the complex cell wall systems found in angiosperms. The only functional evidence linking these expansins to cell wall modification is from Boron et al., 2014, wherein the overexpression of Arabidopsis *EXLA2*, a member of the expansin-like A family, led to slight enlargement of etiolated hypocotyls, accompanied by cell wall thickening and reduced cell wall strength [49].

### 3.3. Gene Co-Expression Network Insights and Correlation with Grain Traits

To determine which temporally regulated expansins are co-expressed with genes implicated in cell wall modification, we used a co-expression network analysis on a dataset of 100 genes derived from a meta-analysis. Co-expression analysis can effectively identify candidate genes involved in specific biological processes and predict the functions of unknown genes [50]. We found that more than half of the differentially expressed expansins in wheat grains (38 out of 72) are clustered in a co-expression module related to cell wall processes (M1, Figure 4). A prior RNA-seq study noted a surge in the expression of genes in the “cell wall” category at 6 DAA [5]. An in-depth examination of the functional domains present within this module’s cell wall genes elucidated that they are chiefly involved in cell wall modification (Figure 5C). This module encompasses expansins with high expression levels at 5 DAA, including *TaEXPB1*, *TaEXPB2*, *TaEXPA9*, *TaEXPA33,* and *TaEXPA1*, supporting the hypothesis that these might be pivotal genes for wheat grain growth.

To support these findings, we validated the expression of these expansins through a qPCR analysis in a separate grain growth experiment and assessed their correlation with grain dimensions and weight. We found a strong negative correlation between expansins *TaEXPA1-A*/*TaEXPB2-B* and grain dimensions (r > 0.9) and weight, indicating their potential involvement in grain growth. Two other expansins from these subfamilies, *TaEXPA3-B1* and *TaEXPB2-B*, also showed a significant correlation with grain weight and size (Appendix A). Previous research corroborates the link between wheat expansins from these subfamilies and cell growth and grain yield. For example, overexpression of wheat *TaEXPA2* in tobacco increased seed production by altering seed size [25], while overexpressing *TaEXPA1* in Arabidopsis boosted germination and growth rate across multiple growth stages [24]. Similarly, the ectopic expression of *TaEXPA6* led to a significant increase in grain size without affecting grain number, resulting in improved yield under field conditions [21]. Overexpression of *TaEXPA2* and *TaEXPB1* had similar effects, causing rapid root elongation, early bolting, and increases in the number of leaves, rosette diameter, and stem length [23]. The findings in Figure 7 suggest that the alpha and beta family expansins identified in this study are promising candidate genes for enhancing wheat grain yield. However, it is important to note that the relationships observed between expansin expression and grain traits require further validation across diverse wheat genotypes. The current study examined a single genotype under field conditions. While meta-analysis provided broader genotypic evidence, an expanded multi-genotype experimental analysis is an important direction for future research to fully confirm the conclusions.

### 3.4. Potential Role and Impact of EXLA Subfamily Expansins on Grain Features

Our analysis of the EXLA subfamily expansins, specifically *TaEXLA3-B* and *TaEXLA2-D*, through quantitative PCR (qPCR), revealed no significant correlation with grain dimensions or weight. This result suggests a divergence in their expression timing in relation to grain growth dynamics (Figure 6). Consequently, this finding indicates that certain expansins may have specialized roles during specific stages of grain development, where their molecular function is required. For instance, while cellularization is generally accepted to be completed by the 7 DAA, endosperm cells continue to proliferate until approximately 12–14 DAA [3]. This proliferation aligns with the observed expression patterns of *TaEXLA2* and *TaEXLA3*, suggesting their possible association with the growth of these cells, which is not seen in the case of alpha and beta expansins. Further, these EXLA subfamily expansins might be related to the growth of specific grain tissues, such as the pericarp. Research indicates the pericarp’s temporal dynamics are distinct from those of the endosperm and embryo [51]. Interestingly, wheat’s maximum pericarp weight is achieved by 12 DAA, correlating with the peak expression of *TaEXLA3* found in our study (10 DAA). Public expression data from the Wheatomics database [52] also reveal that *TaEXLA3* exhibits high expression in the inner pericarp tissue at 12 DAA. This suggests that EXLA expansins may play a role in specific tissue growth during grain development. 

## 4. Materials and Methods

### 4.1. RNA-seq Data Collection and Analysis

The RNA-seq data utilized in this study were carefully curated from the Sequence Read Archive (SRA) hosted by the National Center for Biotechnology Information (NCBI). A rigorous set of selection criteria was enforced to ensure the consistency and reliability of the data. Firstly, only peer-reviewed data were considered to assure the scientific credibility of the sources. To maintain sequencing uniformity across the study, the data included were exclusively generated using the Illumina sequencing platform. The scope of the data was constrained to whole-grain samples, which allowed for a comprehensive investigation into the desired context. Crucially, to enable a detailed examination of gene expression dynamics throughout grain development, the chosen data sets included at least three distinct temporal stages. The datasets utilized in this study were SRA accession numbers PRJNA278920 [26], PRJNA525250 [6], and PRJNA471426 [27].

The Illumina platform was used for sequencing, following a paired-end strategy across all datasets. This uniform approach facilitated more accurate alignment of the sequenced reads. The alignment of reads was performed using the reference transcriptome of *Triticum aestivum* cv. Chinese Spring (IWGSC RefSeq v1.1; [53]) and kallisto software version 0.46.1 [28]. The alignment process adhered to kallisto’s default parameters for index and quantification in paired end mode.

The gene expression levels were also quantified using kallisto [28]. The R package tximport was employed to import raw data from kallisto to DESeq2, ensuring suitable formatting for subsequent analysis [54]. Following quantification, data normalization was carried out using the method implemented in DESeq2 [29].

Differential gene expression analysis was conducted using the statistical methodologies of DESeq2 [29]. Only genes meeting the rigorous criteria of an adjusted *p*-value less than 0.05 and a two-fold change were considered as differentially expressed. The lfcShrink function of the apeglm package was applied for fold change shrinkage in the DESeq2 analysis [55].

Subsequent to the identification of differentially expressed genes, functional analysis was performed. Gene ontology (GO) enrichment was analyzed using the ggprofiler2 package [31], applying the Benjamini–Hochberg method to control the false discovery rate at a 0.05 threshold and a minimum GO term size of 3 was also set for this analysis. A custom gene list was used as the background for this analysis, comprising only genes with a median expression level exceeding 1 transcript per million (tpm) in all grain samples analyzed.

### 4.2. Gene Co-Expression Network Construction

Utilizing the Bionero R package [33], we constructed a gene co-expression network based on the weighted gene co-expression network analysis (WGCNA) algorithm [56]. The network type was set to “signed hybrid” to detect both positive and negative gene correlations. Pearson’s correlation was used to measure the linear correlation between gene expression levels. To estimate gene distances in the network, correlation values were raised to a power β of 16, providing the best fit to a scale-free topology and ensuring the network accurately represents the underlying biological processes. The module merging correlation threshold was set at 0.8, determining when to merge distinct modules based on their dissimilarity.

After network construction, the visualization of the gene co-expression network was conducted using Cytoscape 3.10 [34]. The network was imported into Cytoscape and visualized as an edge-weighted graph, where nodes represented genes and edges represented the co-expression relationships between genes. Edge weights were determined based on the Pearson correlation values obtained from the WGCNA analysis. To focus on the most significant gene co-expression relationships, we implemented a filtering step. Only edges with a Pearson correlation value greater than 0.95 were retained in the network. This stringent threshold allowed us to highlight the strongest gene–gene interactions and to reduce the complexity of the network for more effective visualization and interpretation.

### 4.3. Plant Materials and Sample Collection

This study was conducted during the 2021–2022 growing season at the experimental station of the Universidad Austral de Chile in Valdivia (39°47′0″ S, 73°14′0″ W). *Triticum aestivum* genotype Pantera (INIA) seeds were planted in pots with three seeds per pot. The soil was treated with 150 kg/ha of nitrogen (N) and 300 kg/ha P_2_O_5_ prior to sowing, and an additional 150 kg/ha N was added at tillering. Environmental conditions (temperature, precipitation, and photosynthetically active radiation (PAR)) were monitored by the INIA-Chile-owned Austral Weather Station (http://agromet.inia.cl/ (accessed on 25 June 2023)) (Appendix A). At anthesis, primary spikes were labeled and the first (G1) and second (G2) grains from four central spikelets were selected for sampling. Samples were collected every five days post-anthesis and were flash-frozen in liquid nitrogen for subsequent gene expression analyses. Concurrently, samples for grain weight and grain dimensions were acquired. Each sample type was collected in triplicate for gene expression analysis and in quadruplicate for grain dimension and weight assessment. 

### 4.4. Physiological Measurements of Grains

Upon sampling, grain dimensions (length, width, and area) were measured using a Marvin seed analyzer (Wittenburg, Germany). Following this, the samples were dried at 65 °C for 48 h in a Binder Model FED-720 heater (Tuttlingen, Germany). The grains were then weighed using an electronic balance (Mettler, Germany) to derive the dry grain weight (GW). 

### 4.5. Total RNA Isolation and Gene Expression Analysis by qPCR

For each point of grain dynamics, complete caryopses were pooled from two basal grains of four central spikelets from each spike (nine plants per replicate). The RNA extraction protocol was based on the CTAB method [57], with modifications detailed in [35]. Half a microgram (0.5 µg) of RNA was used to synthesize the first-strand cDNA using 5X All-In-One RT MasterMix (ABM, Inc., Vancouver, BC, Canada), following manufacturer instructions. Gene expression was quantified with Touchdown qPCR assays [36] using Brilliant II SYBR Green QPCR Master Mix (Agilent Technologies, Inc., Santa Clara, CA, USA) and an AriaMx Real-Time PCR System (Agilent Technologies, Inc., Santa Clara, CA, USA). Raw fluorescence data were processed with Real-time PCR Miner 4.0 software [37]. Each qPCR reaction had a total volume of 25 µL, containing 25 ng of cDNA, and each primer was present at 200 µM. The qPCR conditions included one cycle at 95 °C for 10 min; three cycles at 95 °C for 20 s, followed by 66 °C for 10 s, with the temperature decreasing by 3 °C per cycle; and 40 cycles at 95 °C for 20 s, 55 °C for 10 s, and 72 °C for 10 s. Gene-specific forward and reverse primers are listed in Appendix A. The gene ‘*TraesCS4A02G414200*’, a putative ubiquitin-conjugating enzyme, was used as a stable internal control to quantify relative mRNA levels, as previously described for seed development in wheat [38].

### 4.6. Statistical Analysis and Correlation Studies of qPCR Data on Expansin Genes

The analysis of RT-qPCR data for expansin genes was executed using R software (version 4.2.1). The one-way analysis of variance (ANOVA) was used to evaluate the influence of time post-anthesis on the expression of expansin genes. Following ANOVA, pairwise comparisons were conducted employing Tukey’s honestly significant difference (HSD) test. A compact letter display (CLD) was generated from Tukey’s HSD results, providing a succinct visualization of significant differences. The R packages used in this analysis were dplyr, tidyr, ggplot2, multcomp, and multcompView.

Further, correlation analyses were conducted to investigate the association between grain attributes and mRNA expression levels of expansin genes. This was performed using R software (version 4.2.1), with the implementation of ggpubr, ggpmisc, and gridExtra packages. The independent variable was the mRNA expression level of expansin genes, while the dependent variables encompassed four grain attributes: weight, length, width, and area. Each variable pair was depicted on a scatter graph for visual interpretation.

The stat_smooth function from the ggplot2 package, incorporated within ggpubr, was utilized to construct a linear regression model for each pair of variables. The correlation coefficient (R²) and the associated *p*-value were computed and displayed on each graph with the stat_cor function from the ggpubr package. This function offers a straightforward method to calculate and display the correlation coefficient and the *p*-value directly on the graph, easing data interpretation.

## 5. Conclusions

Our findings contribute to the current understanding of the potential role of expansins in wheat grain development, thus laying the groundwork for subsequent investigations into how these proteins affect wheat yield. Notably, our research is not without limitations; our study could not discern the exact functional role of the identified expansins. Future studies should incorporate functional assays to confirm the biological significance of these genes and further delve into their role in wheat grain yield.

Our work expands on the current knowledge of wheat biology by shedding light on the function and importance of expansins in grain development. Future research could explore the suggestion set in this study that the timing of expansins expression picking at 5 or 10 DAA associate with specific grain tissues. Also, we highlight the potential of these expansin genes as genetic markers for enhancing wheat yield. Understanding the precise function of these genes could also enable us to improve wheat grain yield.

In summary, our analysis presents a comprehensive account of the expansin genes potentially contributing to wheat grain growth, thus providing an essential resource for the broader plant biology community. It has opened new questions regarding the specific role of these genes in wheat yield, setting the stage for future in-depth functional investigations.

## Figures and Tables

**Figure 1 plants-12-02868-f001:**
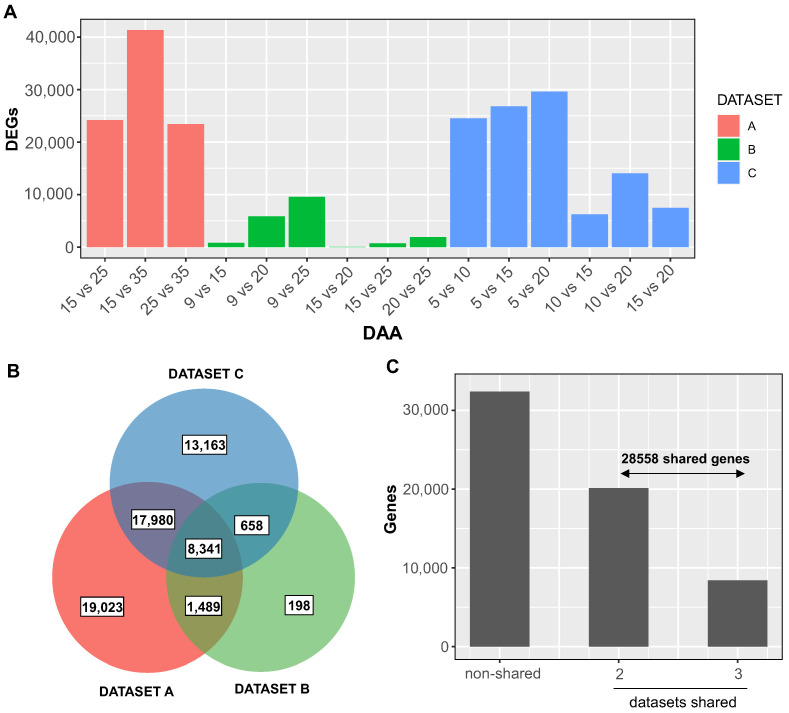
Comparative transcriptomic analysis of RNA-Seq datasets from distinct wheat grain samples. (**A**) Histogram illustrating the differentially expressed genes (DEGs) at various time points (days after anthesis, DAA) during grain growth stages across three distinct datasets (red: dataset A; green: dataset B; blue: dataset C). (**B**) Venn diagram showcasing the overlap of DEGs among the three independent datasets. (**C**) Histogram depicting the total number of unique and shared DEGs across two or three datasets. Dataset A was derived from [26], dataset B was derived from [6], and dataset C was derived from [27]. DEGs were identified using the DESeq2 [29] software, with an adjusted *p*-value less than 0.05 and a two-fold change.

**Figure 2 plants-12-02868-f002:**
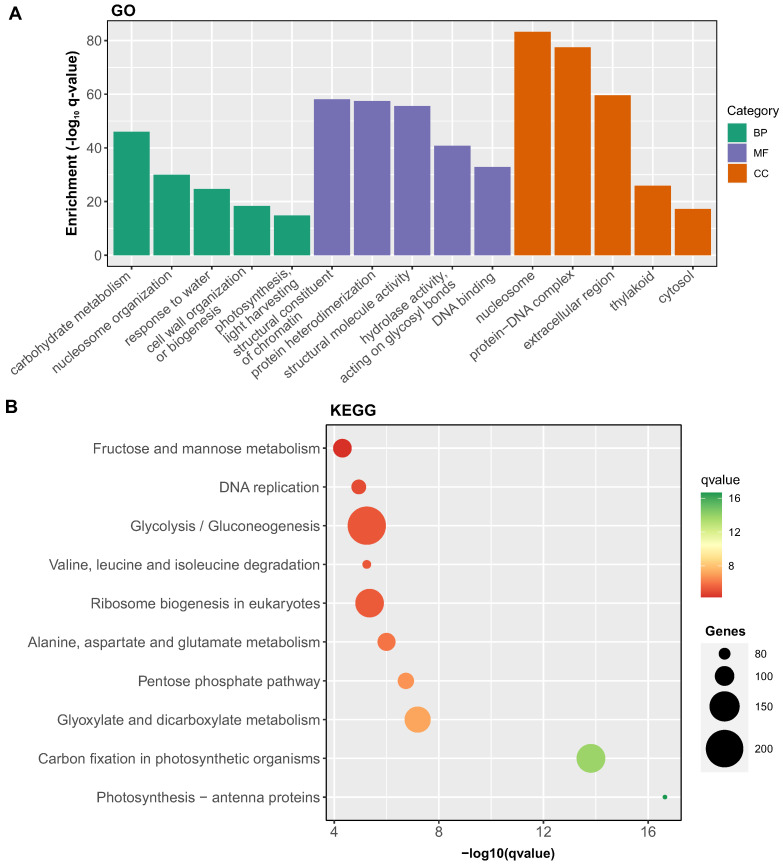
Functional characterization of temporally regulated genes in wheat grain development through gene ontology enrichment analysis. (**A**) Gene ontology (GO) enrichment analysis was performed using the ggprofiler2 package [31], applied to a collection of 28,558 temporally regulated genes that were commonly identified in a minimum of two datasets. To account for multiple hypothesis testing and to control the false discovery rate, the analysis adopted the Benjamini–Hochberg method, applying a significance threshold of 0.05. The figure highlights the five most significantly over-represented GO terms within each of the three categories: biological process (green), molecular function (purple), and cellular component (red). (**B**) Enriched KEGG pathways among differentially expressed genes during wheat grain development. KEGG pathway enrichment analysis was conducted using the ClusterProfiler R package [32]. The top 10 significantly enriched pathways are displayed, with the pathway description on the *y*-axis and the −log10 transformed adjusted *p*-value (q-value) on the *x*-axis. Dot size corresponds to the number of differentially expressed genes annotated for each pathway.

**Figure 3 plants-12-02868-f003:**
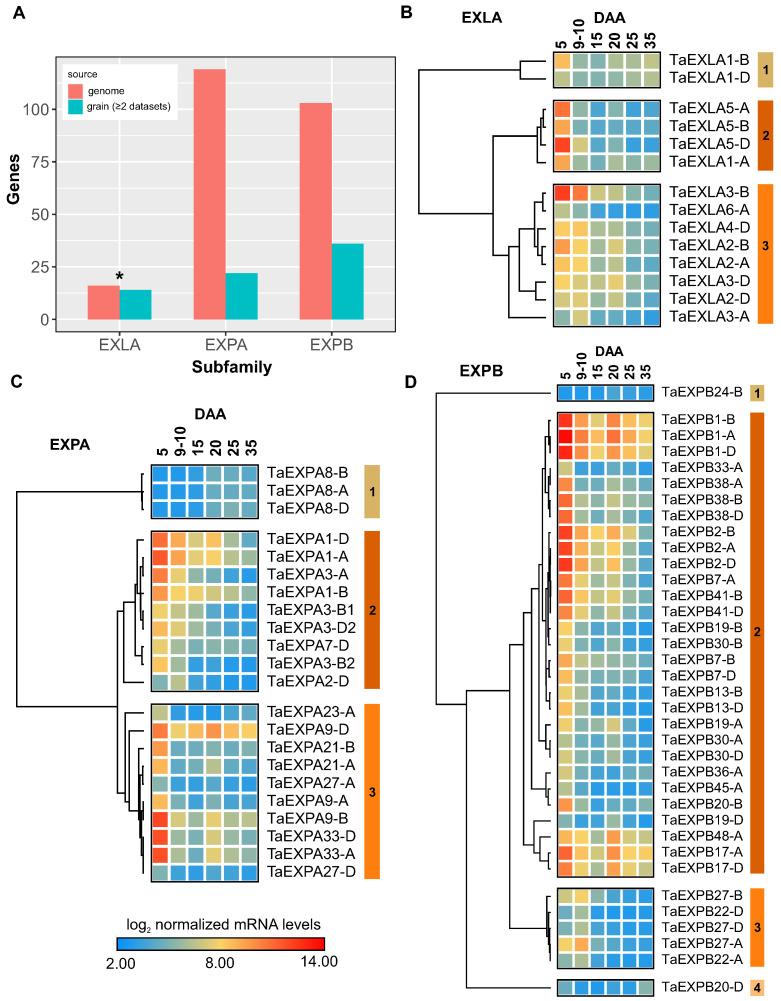
Temporal dynamics and subfamily distribution of expansin genes during grain development. Panel (**A**) illustrates a histogram contrasting the total count of expansin genes present in the genome with those detected in our comprehensive analysis of grain development. Asterisk indicates significant overrepresentation (*p*-value < 0.01). Panels (**B**–**D**) exhibit hierarchical clustering analyses for the EXLA, EXPA, and EXPB subfamilies, respectively. The hierarchical clustering of expansin genes was conducted employing Morpheus software (https://software.broadinstitute.org/morpheus (accessed on 25 January 2023)), utilizing a 1-Pearson correlation value as the metric, with the average linkage method derived from DESeq2 [29] normalized expression levels on a log2 scale. To simplify the data visualization, only the mean expression levels across all samples from the same days after anthesis (DAA) are displayed.

**Figure 4 plants-12-02868-f004:**
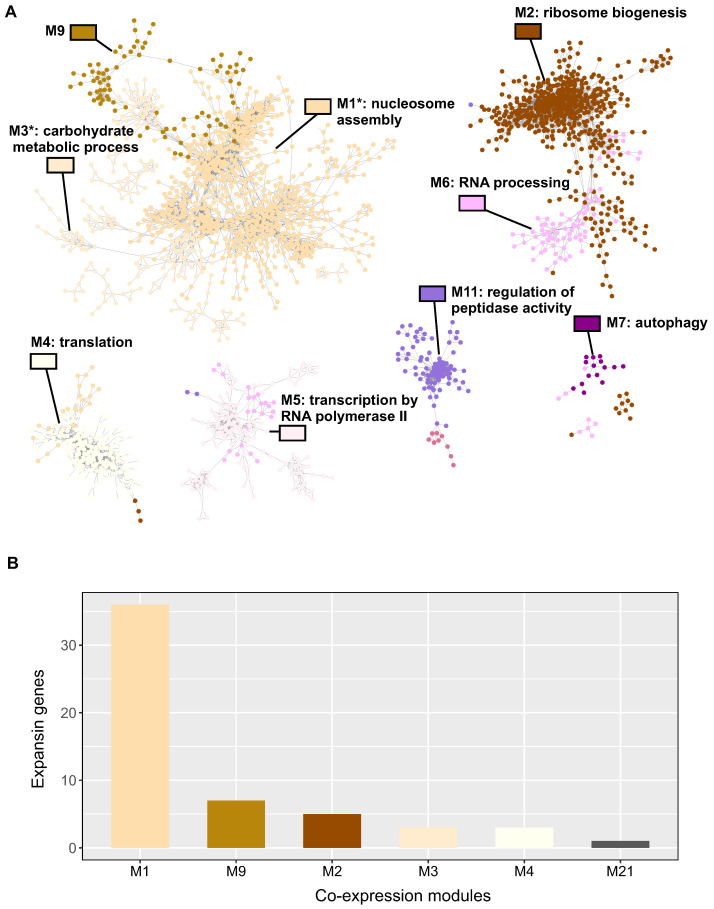
Gene co-expression network analysis of differentially expressed genes in grain development identifies two specific modules associated with cell wall function. (**A**) Co-expression network showcasing genes of the same color for each identified co-expression module using BioNERO r package [33], highlighting the most enriched gene ontology (GO) term based on the over-representation analysis conducted with the ggprofiler2 r package [31]. Modules marked with asterisks represent genes associated with cell wall activity. For ease of network visualization, only interactions with a Pearson correlation of 0.95 or above are shown; hence, modules containing genes below this correlation threshold are not displayed. (**B**) Distribution of expansin genes within the co-expression network modules, emphasizing module M1 in terms of the number of expansins present. Cytoscape 3.10 [34] was employed to generate the visualization of the gene co-expression network.

**Figure 5 plants-12-02868-f005:**
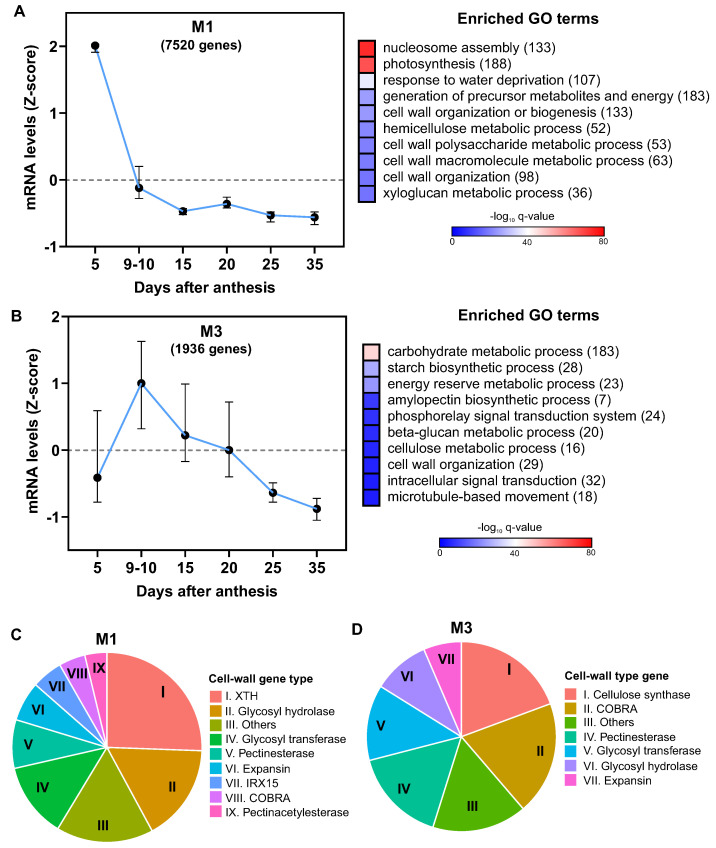
Gene expression patterns and related biological functions in two identified modules associated with cell wall organization. Displayed are the expression profiles of differentially expressed genes (DEGs) within Module M1 (**A**) and Module M3 (**B**), obtained across various grain samples analyzed in this study. For each gene, average mRNA levels at the same time point across all samples were normalized using Z-score. Accompanying charts illustrate the enriched gene ontology (GO) terms with their corresponding enrichment q-values depicted in a −log10 scale, color-coded for clarity. The quantity of DEGs annotated for each GO term is indicated in parentheses. Pie charts present the distribution of gene families related to cell wall within Modules M1 (**C**) and M3 (**D**). This is based on the InterPro domain functional annotation analysis of genes linked to the commonly enriched GO term “cell wall organization or biogenesis” observed in both modules. For this analysis, data were sourced from the Ensembl Plants database (https://plants.ensembl.org/ (accessed on 30 February 2023)).

**Figure 6 plants-12-02868-f006:**
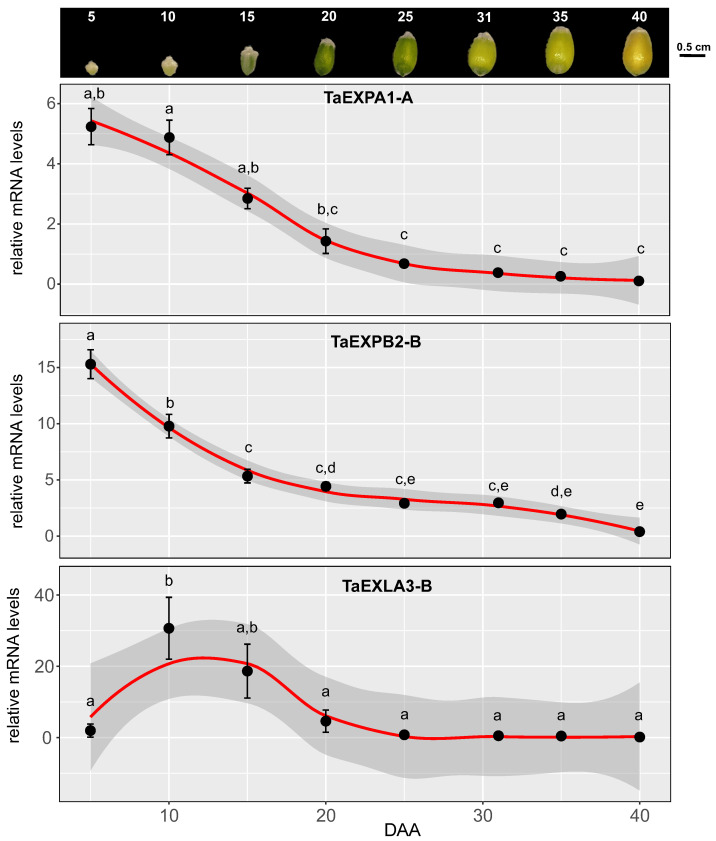
Quantitative analysis of selected expansin genes during wheat grain expansion. This figure demonstrates the use of quantitative real-time PCR (qRT-PCR) to validate the expression of selected expansin genes during the expansion of wheat grains. The images at the top section represent wheat grains collected at eight distinct developmental stages (5, 10, 15, 20, 25, 31, 35, and 40 days after anthesis, DAA) during the 2021–2022 growing season in an experiment described in the Materials and Methods section. These samples, collected in triplicate, were immediately preserved in liquid nitrogen for subsequent RNA isolation [35]. The touchdown qPCR assays [36], performed with Brilliant II SYBR Green QPCR Master Mix, quantified gene expression. The raw fluorescence data were subsequently analyzed using Real-time PCR Miner software [37]. We used the ‘*TraesCS4A02G414200*’ gene, a presumed ubiquitin-conjugating enzyme, as a stable internal control for quantifying relative mRNA levels, as previously applied to wheat seed development [38]. Statistical analyses of the qPCR data involved R (version 4.2.1) and a one-way analysis of variance (ANOVA). Post-hoc pairwise comparisons were executed using Tukey’s honestly significant difference (HSD) test. Results led to the assignment of different letters to significantly different means, with significance defined as a *p*-value < 0.05.

**Figure 7 plants-12-02868-f007:**
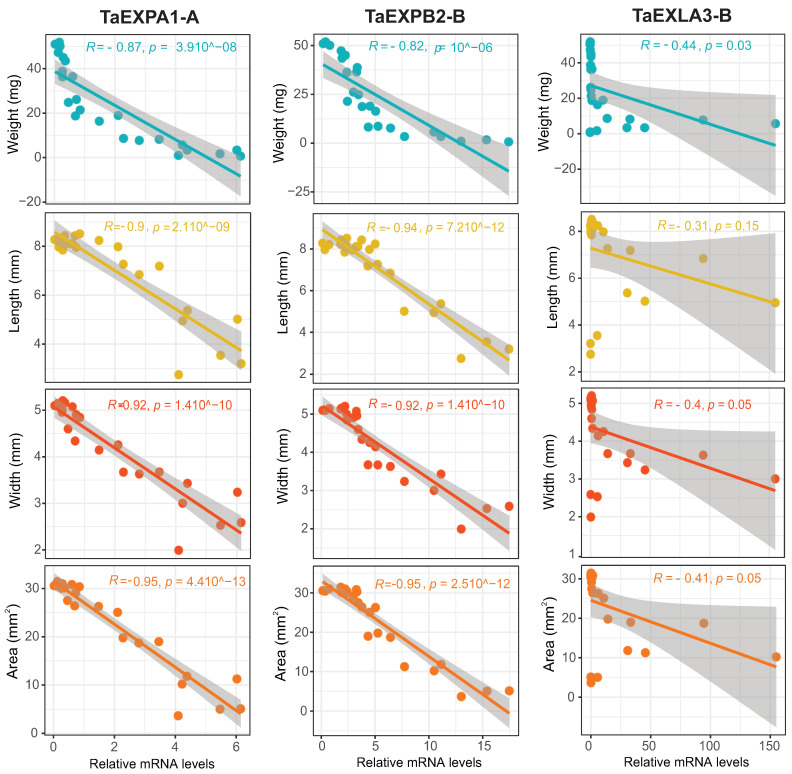
Scatter plots depicting correlations between mRNA level of selected expansins and grain parameters. Data points represent grain weight, length, width, and area against the mRNA level of *TaEXPA1-A*, *TaEXPB2-B*, and *TaEXLA-3B*. The fitted linear regression model is displayed on each scatter plot, along with the correlation coefficient (R²) and *p*-value. *TaEXPA1-A* and *TaEXPB2-B* show a significant negative correlation with grain growth parameters (*p*-value < 0.01, R < −0.8), whereas *TaEXLA-3B* does not exhibit any significant correlation. The data analysis and visualization were performed using R (version 4.2.1) with the ggpubr, ggpmisc, and gridExtra packages.

**Table 1 plants-12-02868-t001:** Summary of RNA-seq datasets and sample details.

Name	Accession Number	Database	Samples	Tissue	DAA	Growth Condition	Genotype	Replicates	References
Dataset A	PRJNA278920	SRA	84	Whole grain	15, 25, 35	Glasshouse	Five near-isogenic lines and four parent varieties	2	Barrero et al., 2015 [26]
Dataset B	PRJNA525250	SRA	8	Whole grain	5, 10, 15, 20	Field	Wheat cultivar Xiaoyan-6	2	Chi et al., 2019 [6]
Dataset C	PRJNA471426	SRA	8	Whole grain	9, 15, 20, 25	Field	Wheat cultivar Shumai 482	2	Zhong et al., 2021 [27]

## Data Availability

The RNA-Seq datasets analyzed during this study are available in the NCBI Sequence Read Archive (SRA) repository, accession numbers PRJNA278920, PRJNA525250, and PRJNA471426. All other data generated during this study are included in this published article and its Appendix A.

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
