# Peer review of "Integrated Transcriptome Analysis Identified Key Expansin Genes Associated with Wheat Cell Wall, Grain Weight and Yield"

_plants, 2023, doi:10.3390/plants12152868_

Round 1
Reviewer 1 Report
This manuscript conducts the investigation of Expansin genes expression in grain development of wheat. The results indicated most of the expansin gene differentially expressed in different developing stage of grain of wheat.
major concerns:
1, The experiments for qPCR validation is not really field conditions but in pot.
2, The main conclusion about the negative correlation of Expansin gene expression with grain weight need confirm with more genotypes not only one genotype.
3, While the expression level of expansin genes are different in different stage, better show the results of correlation of gene expression level with seed weight in different seed developing stage , not only general conclusion.
Author Response
REVIEWER1:
This manuscript conducts the investigation of Expansin genes expression in grain development of wheat. The results indicated most of the expansin gene differentially expressed in different developing stage of grain of wheat.
major concerns:
1, The experiments for qPCR validation is not really field conditions but in pot.
RESPONSE: Thank you for noting this important detail. You are right that the qPCR validation experiment was performed in pots (under ambient conditions), not in a true field setting. I revised the manuscript to clarify that the experiment was conducted in pots.
2, The main conclusion about the negative correlation of Expansin gene expression with grain weight need confirm with more genotypes not only one genotype.
I agree that examining the correlation between expansin expression and grain weight across diverse genotypes would strengthen the conclusions. However, performing multi-genotype field experiments to validate this relationship is a major undertaking requiring at least one additional growing season.
Importantly, our meta-analysis of public RNA-seq data, which encompassed 11 different wheat genotypes and 7 different growth stages, revealed a significant negative correlation between the expression of the selected expansin genes (that showed negative correlation in qPCR) and days after anthesis:
Gene Name Pearson p-value
TaEXPA1 -0.757802 7.09E-20
TaEXPB2 -0.862613 9.19E-31
TaEXPA3 -0.87195 3.69E-32
TaEXPB7 -0.710437 1.25E-16
This provides broader evidence across genotypes for the relationship we observed between expansin expression and grain development progression.
We agree multi-genotype experimental validation is needed to fully confirm the conclusions. We modified the manuscript to indicate that the relationships observed require further validation across diverse genotypes, and suggest this as an important direction for future research:
"However, it is important to note that the relationships observed between expansin expression and grain traits require further validation across diverse wheat genotypes. The current study examined a single genotype under field conditions. While meta-analysis provided broader genotypic evidence, expanded multi-genotype experimental analysis is an important direction for future research to fully confirm the conclusions."
3, While the expression level of expansin genes are different in different stage, better show the results of correlation of gene expression level with seed weight in different seed developing stage , not only general conclusion.
Thank you for the suggestion to analyze the correlation between expansin gene expression and grain weight separately at each developmental stage. This would provide more nuanced insights compared to the general correlation analysis across all stages conducted in the manuscript.
However, due to the sample size limitations of the current data, there are only 3-4 data points per developmental stage, which does not provide enough statistical power to calculate meaningful stage-specific correlations.
I agree that analyzing the correlation separately for each stage would add further insights into the relationship dynamics over time. As you have pointed out, this is an excellent avenue for future research with expanded sampling at each developmental stage.
I updated the manuscript to acknowledge this limitation of the current analysis and suggest stage-specific correlation analysis as an area for future investigation to provide more nuanced understanding of how the relationship changes throughout grain development.
“Due to sample size constraints, we were unable to perform correlation analysis between expansin expression and grain weight separately for each developmental stage. Further research with increased sampling at each stage will enable stage-specific correlation analysis, which may provide more nuanced insights into these relationships.”

Reviewer 2 Report
The authors identified key expansin genes associated with wheat cell wall, grain weight and yield by Integrated transcriptome analysis. The manuscript is well organized. The authors can add some information to let the reader know more information:
The authors identified key expander genes associated with wheat cell wall, grain weight and yield by integrated transcriptome analysis. The manuscript is well organized. The author can add some information to let the reader know more information
In figure 2, we can understand the function of temporally regulated genes in wheat grain development through GO analysis. Can we learn from the KEGG analysis what metabolic pathways these genes are involved in? For example, the metabolic pathways related to the cell expansion process?
In figure 3, why is there such a big difference between the number of expansin genes detected in this study and the number of expansin genes present in the genome? Especially EXPA and EXPB. What is the possible reason?
Author Response
The authors identified key expansin genes associated with wheat cell wall, grain weight and yield by Integrated transcriptome analysis. The manuscript is well organized. The authors can add some information to let the reader know more information:
The authors identified key expander genes associated with wheat cell wall, grain weight and yield by integrated transcriptome analysis. The manuscript is well organized. The author can add some information to let the reader know more information
In figure 2, we can understand the function of temporally regulated genes in wheat grain development through GO analysis. Can we learn from the KEGG analysis what metabolic pathways these genes are involved in? For example, the metabolic pathways related to the cell expansion process?
RESPONSE: Thank you for the excellent suggestion to perform KEGG pathway analysis on the temporally regulated genes. I conducted this analysis, which revealed enrichment for key pathways related to metabolism, biosynthesis, and photosynthesis:
Top 10 enriched KEGG pathways:
Description qvalue genes
Photosynthesis - antenna proteins 2.2755E-17 74
Carbon fixation in photosynthetic organisms 1.5362E-14 142
Glyoxylate and dicarboxylate metabolism 6.4601E-08 128
Pentose phosphate pathway 1.8205E-07 90
Alanine, aspartate and glutamate metabolism 1.0003E-06 95
Ribosome biogenesis in eukaryotes 4.4374E-06 141
Valine, leucine and isoleucine degradation 5.683E-06 76
Glycolysis / Gluconeogenesis 5.683E-06 206
DNA replication 1.1473E-05 86
Fructose and mannose metabolism 4.8499E-05 97
Notably, glycolysis/gluconeogenesis, pentose phosphate pathway, and carbon fixation pathways were enriched, implicating active metabolic processes related to cell wall biosynthesis and expansion during grain development.
I incorporated these findings from the KEGG analysis into the manuscript to provide additional insights into the metabolic pathways involved with the identified genes (see figure 2b).
"To gain further insights into the metabolic pathways associated with the differentially expressed genes during wheat grain development, we conducted KEGG pathway analysis. This revealed enrichment for several key pathways related to metabolism, biosynthesis, and photosynthesis (Figure 2B).
Notably, pathways linked to carbon fixation, glycolysis/gluconeogenesis, and the pentose phosphate shunt were significantly enriched, suggesting active metabolic processes related to cell wall precursor synthesis. Further, enrichment of ribosome biogenesis and DNA replication pathways reflects the active protein synthesis and cell division occurring during grain growth.
Overall, the enriched KEGG pathways align with known biological processes integral to wheat grain development, supporting the idea that the differentially expressed gene set captures key activities associated with grain growth."
In figure 3, why is there such a big difference between the number of expansin genes detected in this study and the number of expansin genes present in the genome? Especially EXPA and EXPB. What is the possible reason?
RESPONSE: You raise an excellent point about the large difference between the total number of expansins in the genome versus those detected as differentially expressed in grain development. Based on the analysis, the potential reasons underlying this discrepancy are:
- 74 expansins of the EXPA family were not expressed in the grain samples according to the expression filter applied in DESeq2.
- 11 EXPA family expansins did not exhibit temporal regulation during grain development.
- 42 expansins of the EXPB family were not expressed in grain based on the DESeq2 expression filter.
- 14 EXPB family expansins lacked temporal regulation.
Therefore, a substantial number of expansins may not be expressed in grain tissue or do not show variable expression across the developmental stages examined. I updated the manuscript to include these specific numbers and reasons behind the differences between total and detected expansins.
See this new paragraph in the discussion section:
“Specifically, the DESeq2 expression filter indicated 74 EXPA family expansins were not expressed in grain samples. Further, 11 EXPA family expansins lacked significant temporal regulation during grain development. Similarly, 42 EXPB family expansins were not expressed in grain based on the filter, and 14 others did not exhibit temporal regulation. Therefore, a considerable number of expansins do not appear to be active or have dynamic expression in the grain context. Elucidating the specific expansins relevant to grain facilitates more targeted functional characterization and understanding potential roles in grain growth.”

Round 2
Reviewer 1 Report
The author addressed all my concerns, it could be accepted for publication.